# Serum Urate Levels and Ultrasound Characteristics of Carotid Atherosclerosis across Obesity Phenotypes

**DOI:** 10.3390/biomedicines11071897

**Published:** 2023-07-04

**Authors:** Daniela Efremova, Natalia Ciobanu, Danu Glavan, Pavel Leahu, Renata Racila, Tatiana Bălănuță, Alexandru Matei, Maria Vasilieva, Cristina Cheptea, Paula Bîtcă, Cristina Damian, Ana Bondarciuc, Irina Bejenari, Adelina Cojocaru, Diana Manea, Mihail Ciocanu, Eremei Zota, Dumitru Ciolac, Stanislav A. Groppa

**Affiliations:** 1Department of Neurology, Institute of Emergency Medicine, 2004 Chisinau, Moldova; daniela.efremova88@gmail.com (D.E.); natalia.ciobanu@usmf.md (N.C.); glavandan@gmail.com (D.G.); pavel.leahu@usmf.md (P.L.); renata.racila@gmail.com (R.R.); tatianabalanutsa@gmail.com (T.B.); dadmatei@yahoo.com (A.M.); fbi-miv@mail.ru (M.V.); cristinaburlacu89@yahoo.com (C.C.); paulinabitca@gmail.ru (P.B.); kristina.damian@gmail.com (C.D.); ana_bondarciuc@mail.ru (A.B.); ira.bejenari.93@mail.ru (I.B.); casianadelina@gmail.com (A.C.); dimanea@gmail.com (D.M.); mihai.ciocanu@gmail.com (M.C.); ezotajr@gmail.com (E.Z.); 2Department of Neurology, Nicolae Testemitanu State University of Medicine and Pharmacy, 2004 Chisinau, Moldova

**Keywords:** serum urate, intima-media thickness, carotid plaques, obesity phenotype

## Abstract

Background: Existing evidence suggests a close link among high levels of serum urate (SU), obesity and carotid atherosclerosis. The aim of the present study was to evaluate the interrelations between SU levels and carotid atherosclerosis in subjects with different obesity phenotypes. Methods: In this study, a total of 2076 subjects (mean age 48.1 ± 13.1 years; 1307 women) were recruited: 59 with general obesity, 616 with central obesity, 715 with mixed (general–central) obesity and 686 non-obese. Anthropometric measurements, vascular risk factors, blood biochemistry analysis (including SU levels), and carotid ultrasound were performed. Ultrasound assessment included evaluation of intima-media thickness (IMT) and plaque characteristics, including number, total area and type (vulnerable vs. stable) of plaques. Results: After adjustment for potential confounders, the highest levels of SU were observed in subjects with mixed obesity, followed by subjects with central obesity, general obesity and the non-obese (309.4 ± 82.2 vs. 301.2 ± 73.1 vs. 272.9 ± 61.8 vs. 234.2 ± 59.8 μmol/L, respectively; F = 149.2, post hoc *p* < 0.001). Similarly, subjects with mixed and central obesity presented higher values of IMT compared to subjects with general obesity and the non-obese (0.68 ± 0.16 vs. 0.67 ± 0.16 vs. 0.62 ± 0.14 vs. 0.57 ± 0.13 mm, respectively; F = 54.2, post hoc *p* < 0.001). No difference in number, total area and type of plaques among obesity groups were attested (all *p* > 0.05). Significantly higher IMT values were observed in subjects with increased SU levels compared to subjects with normal SU levels (0.70 ± 0.10 vs. 0.62 ± 0.14 mm, *p* = 0.02) only within the central obesity group. Increasing levels of SU were associated with a higher frequency of increased IMT only in subjects with central obesity (OR 1.033, 95% CI 1.025–1.041). Similarly, SU levels yielded a satisfactory performance in detecting subjects with increased IMT (AUC 0.65, 95% CI 0.50–0.73, subjects with carotid plaques (0.62, 95% CI 0.55–0.68) and subjects with vulnerable plaque types (0.68, 0.59–0.76) only within the central obesity group. Conclusions: Among the studied obesity types, the association between SU levels and markers of carotid atherosclerosis was of particular significance in subjects with central obesity.

## 1. Introduction

Cardiovascular and cerebrovascular diseases are the leading cause of death globally [1]. A spectrum of various modifiable risk factors, such as arterial hypertension, diabetes, obesity, physical inactivity, tobacco use, alcohol consumption, and dyslipidemia, were recognized as major vascular risk factors. Nevertheless, other less common risk factors are directly or indirectly associated with cerebrovascular diseases. Among these, elevated levels of serum urate (SU) are emerging as an important risk factor for cerebrovascular diseases [2,3,4]. The specific role of SU in cerebrovascular diseases, as an independent causal factor, a mediator or a marker of vascular comorbidities, is still the matter of debate. Despite existing controversies, increased SU levels have been independently associated with many vascular risk factors, including hypertension, obesity, diabetes mellitus, dyslipidemia, atherosclerosis and metabolic syndrome [3,5]. Depicting the role of SU in different vascular comorbidities may improve the risk stratification for cerebrovascular diseases.

Obesity is a major global health problem and poses a significant burden on healthcare systems. Obesity is known to be strongly related to a higher prevalence of hypertension, diabetes and metabolic syndrome [6] and to be significantly associated with an increased risk of cerebrovascular diseases [7]. In addition, obesity was shown to be associated with SU levels. In particular, a close relationship between SU and body mass index (BMI) and waist circumference (WC) was documented, with higher SU levels being detected in subjects with higher BMI [8,9,10] and higher WC values [11,12]. It has been postulated that abdominal obesity in particular is one of the main contributors of elevated SU levels in the general population [5,13]. These data indicate the existence of potential implications of obesity phenotypes in determining the SU levels and associated vascular risk factors.

Carotid atherosclerosis is a common pathological substrate of cerebrovascular diseases, being one of the major causes of ischemic stroke [14,15]. Currently, carotid intima–media thickness (IMT) and carotid plaques are widely used as non-invasive markers for carotid atherosclerotic disease and are directly associated with an increased risk of cerebrovascular diseases [16,17,18]. Moreover, the morphology of carotid plaques as assessed by carotid ultrasound is closely related to the risk of stroke—the less echogenic the plaque is (i.e., the more vulnerable), the higher the risk of stroke [19,20]. To date, numerous studies have demonstrated that higher SU levels are associated with higher values of IMT in healthy as well as in diseased (e.g., hypertensive or diabetic) populations [21]. Similarly, higher SU levels were associated with a higher occurrence of carotid plaques [22] and with a higher prevalence of vulnerable plaques [23]. However, little is known about the possible differential effects of SU on carotid atherosclerosis in subjects with different types of obesity.

Given that obesity is associated with both high SU levels and carotid atherosclerotic disease, it is essential to characterize SU levels and markers of carotid atherosclerosis across different obesity types. Thus, in the present study, we aimed to explore the interrelations between SU levels and carotid atherosclerosis (IMT and carotid plaque characteristics) in subjects with general, central and mixed (general–central) types of obesity.

## 2. Materials and Methods

### 2.1. Study Subjects

This cross-sectional study was part of the National State Program on Stroke Risk Factors initiated in 2015 that aimed to characterize the risk factors for stroke in the population of the Republic of Moldova. The inclusion criteria were as follows: age ≥ 18 years and signed informed consent. Subjects aged <18 years, a history of stroke, the presence of a major systemic disease (e.g., cancer, heart, liver or renal insufficiency), and failure to sign the informed consent served as exclusion criteria. Screening activities within the national program were performed in two stages, as described in [24,25]. First, local general practitioners prepared the lists of subjects in their circumscription, and all members aged ≥18 years were recruited for participation. In the second stage, subjects were interviewed and clinically examined in accordance with the study protocol for the evaluation of risk factors for stroke. Demographic data, previous medical history, concomitant medication and behavioral factors were determined using a predesigned questionnaire. Among the concomitant medication potentially affecting the levels of SU, only data on aspirin administration were available for the included subjects. All interviews as well as physical and instrumental evaluations were carried out by a well-trained team of physicians (from neurology and internal medicine disciplines). In total, 2076 subjects were evaluated and included into the study.

### 2.2. Physical Evaluation

For each subject, the following measures were performed: body weight and height for the BMI, WC, and blood pressure (BP). Body mass index was calculated as weight divided by the square of the body height. General obesity was defined as BMI ≥ 30 kg/m^2^ [26]. Waist circumference was measured by wrapping the tape around the midway between the lowest border of the ribs and iliac crest in the horizontal plane at the end of a normal breath. Central obesity was defined as an WC of >80 cm in women and >94 cm in men according to World Health Organization [26]. Mixed obesity was defined as coexistence of both general and central obesity in the same subject.

Blood pressure (BP) was measured on both extremities in clinostatic and orthostatic positions. Hypertension was estimated as a self-reported history of arterial hypertension or taking antihypertensive medication [27].

### 2.3. Laboratory Examination

Venous blood samples were collected via venipuncture from each subject after overnight fasting for evaluation of SU levels, serum fasting blood glucose (FBG), and the lipid profile (total cholesterol (TC), triglycerides (TG), high-density lipoprotein cholesterol (HDL-C) and low-density lipoprotein cholesterol (LDL-C)), according to established protocols. High TC was estimated as ≥5.0 mmol/L, high LDL-C as ≥4.16 mmol/L, and low HDL-C as ≤1.04 mmol/L. Dyslipidemia was defined as elevated TC or LDL-C levels, or low levels of HDL-C [28]. Regarding SU, there is no universally accepted definition for high SU levels. In our study, high SU levels were defined as serum concentrations > 416.4 μmol/L (or 7 mg/dL) in males and >356.9 μmol/L (or 6 mg/dL) in females. These cut-off values were based on generally accepted values used in clinical laboratories and in previous studies [29,30].

### 2.4. Ultrasound Assessment

In order to evaluate the IMT and plaque morphology, Doppler-Duplex ultrasound was used to assess the extracranial portions of the carotid arteries–the common carotid artery (CCA), the bifurcation, the internal carotid artery (ICA), and the external carotid artery (ECA) [31]. After the bifurcation of the CCA was confirmed, the carotid IMT was measured 2 cm proximal to the dilatation of the carotid bulb [32]. Values of IMT ≥ 0.9 mm were considered as increased IMT. Atherosclerotic plaque was defined as a focal structure that penetrates into the arterial lumen with 0.5 mm or 50% of the adjacent IMT or has a thickness of ≥1.5 mm [33]. Ultrasound characterization of plaques was performed bilaterally on 3 segments: the CCA, the bifurcation and the ICA. The morphology of plaques was described based on the assessment of three main features: echogenicity, homogeneity and surface. Area (in mm^2^) of the plaques was quantified by measuring the cross-sectional area of each plaque in a longitudinal view (in which the plaque was largest), freezing the frame, and tracing around the plaque perimeter with a cursor on the screen [34]. Distinction of plaque types was performed according to the modified Gray–Weale classification: type 1—uniformly an- or hypoechogenic; type 2—predominantly hypoechogenic; type 3—predominantly hyperechogenic; type 4—uniformly hyperechogenic; and type 5—unclassified due to calcification [35]. As previously mentioned, the morphological characteristics of the plaques are related to the risk of stroke—the lower the echogenicity of the plaque, the higher the risk of stroke [19]. In this regard, atherosclerotic plaques of types 1 and 2 are considered vulnerable (or unstable) plaques, with a high risk of stroke, while types 3, 4 and 5 are considered stable plaques, with a lower risk of stroke [36]. Eventually, for the subsequent analysis, the following ultrasound parameters were considered: IMT (mean of right and left sides), number, and type and total area of the plaques (assessed on both CCA and ICA).

### 2.5. Statistical Analysis

Gaussian distribution of the variables was assessed by evaluating the histograms and Shapiro–Wilk test. Variables are presented as means, medians or proportions where appropriate. Differences between the means of continuous variables were assessed via a two-sample *t*-test or one-way analysis of variance (ANOVA) and Pearson’s χ^2^ for categorical variables.

Between-group differences in laboratory and ultrasound parameters were assessed by employing general linear models (GLMs). The models were adjusted for age, sex, concomitant medication (aspirin), creatinine and other cardiovascular risk factors (i.e., arterial hypertension, diabetes mellitus, smoking, total C, LDL-C and HDL-C), where appropriate.

Binary logistic regression was employed to estimate the association between SU levels (as the independent variable) and (i) occurrence of increased IMT, (ii) occurrence of carotid plaques and (iii) occurrence of vulnerable carotid plaques (as dependent variables) in each obesity group. The models were adjusted for age, sex, concomitant medication (aspirin), creatinine and other cardiovascular risk factors. Odds ratios (ORs) with confidence intervals (CIs) were calculated.

A receiver operating characteristic (ROC) analysis was performed to assess the performance of SU levels to detect (i) subjects with increased IMT, (ii) subjects with carotid plaques and (iii) subjects with vulnerable carotid plaques (types 1 and 2) in each obesity group. For each model, the area under the curve (AUC) and CIs were calculated. The performance of models was estimated as excellent (AUC 0.9–1.0), very good (AUC 0.8–0.9), good (AUC 0.7–0.8), satisfactory (AUC 0.6–0.7) and unsatisfactory (AUC 0.5–0.6).

All analyses were carried out using SPSS (version 23.0; IBM, Armonk, NY, USA).

## 3. Results

### 3.1. General Characteristics

In this study, 2076 (1307 women) subjects with a mean age of 48.1 ± 13.1 years were included. Among these, 1390 were with different types of obesity (either general, central or mixed) and 686 were non-obese (Table 1). Non-obese subjects were younger compared to obese subjects (ANOVA, F = 71.8, all post hoc *p* < 0.001). General and mixed obesity groups contained mainly female subjects, while the central obesity group contained mainly male subjects (χ^2^ = 261.5, *p* < 0.001). Subjects with general and mixed obesity presented the highest values of BMI (F = 145.4, all post hoc *p* < 0.001). Subjects with mixed obesity displayed the highest values of abdominal circumference compared to all other three groups (F = 135.1, all post hoc *p* < 0.001). Among the obesity groups, systolic and diastolic BP did not significantly differ (all post hoc *p* > 0.05) but was higher compared to the non-obese group (all post hoc *p* < 0.001). Arterial hypertension was more frequent in subjects with general and mixed obesity compared to subjects with central obesity (χ^2^ = 89.9, *p* < 0.001). The highest frequency of smoking (25%) was among subjects with central obesity compared to subjects with general and mixed obesity (χ^2^ = 130.6, *p* < 0.001), while diabetes mellitus was more frequent in subjects with mixed obesity (χ^2^ = 67.3, *p* < 0.001). The frequency of dyslipidemia was similar among subjects with different types of obesity (χ^2^ = 2.2, *p* = 0.13). Subjects with central obesity showed higher values of triglycerides (post hoc *p* < 0.001) and lower values of HDL-C (post hoc *p* < 0.001) compared to subjects with mixed obesity. No significant differences in TC and LDL-C were attested among obesity types (all post hoc *p* > 0.05). Creatinine levels were similar among the obesity groups (all post hoc *p* > 0.05) but higher compared to the non-obese group (all post hoc *p* < 0.001). The highest prevalence of aspirin consumption was observed in subjects with mixed and general obesity compared to subjects with central obesity (32 vs. 27 vs. 15%, respectively; χ^2^ = 50.3, *p* < 0.001). Subject characteristics are presented in Table 1.

### 3.2. Serum Urate Levels

Levels of SU were significantly different among subjects with different obesity types (F = 125.1, all post hoc *p* < 0.001), with highest levels being detected in subjects with central and mixed obesity (Table 1, Figure 1). Within all the studied population, increased levels of SU were identified in 192 (9.2%) subjects, with a higher frequency in all obesity groups compared to non-obese subjects (χ^2^ = 89.3, *p* < 0.001). Within obesity groups, the frequency of increased SU levels was the highest in subjects with mixed obesity compared to subjects with general and central obesity (χ^2^ = 28.7, *p* < 0.001), for which the frequency was similar (χ^2^ = 0.4, *p* = 0.52) (Table 1).

### 3.3. Ultrasound Characteristics

The highest values of IMT were attested in subjects with central and mixed types of obesity compared to subjects with general obesity and the non-obese (F = 55.7, *p* < 0.001) (Figure 1, Table 2). The frequency of increased IMT did not differ across the subjects with either type of obesity (χ^2^ = 3.1, *p* = 0.21). The frequency (χ^2^ = 2.9, *p* = 0.23), number (F = 1.89, *p* = 0.12) and area (F = 2.13, *p* = 0.17) of the plaques were similar among the obesity groups but higher compared to the non-obese (all *p* < 0.001) (Table 2, Appendix A). The type of plaque had a comparable distribution among all the subject groups (χ^2^ = 1.1, *p* = 0.77), with stable types of plaque (i.e., types 3, 4 and 5) predominating in all the groups.

### 3.4. Serum Urate Levels and Carotid Atherosclerosis

Next, we compared the IMT, number and area of the plaques between subjects with increased and normal SU levels in each studied group. Subjects of either group with increased SU levels tended to have higher IMT estimates, a higher number of plaques and a greater plaque area compared to subjects with normal SU levels. However, the difference was not statistically significant, except for the subjects with central obesity, in which higher IMT values were observed in subjects with increased SU levels (0.70 ± 0.10 vs. 0.62 ± 0.14 mm, *p* = 0.02) (Figure 2 and Appendix A).

Table 3 shows the relationship between SU levels and ultrasound markers of carotid atherosclerosis estimated with logistic regression models. Increasing levels of SU were associated with a higher frequency of increased IMT only in subjects with central obesity (OR 1.027, 95% CI 1.021–1.038). Interestingly, SU levels were inversely associated with the frequency of carotid plaques in the non-obese (OR 0.992, 95% CI 0.988–0.996). No other associations between SU levels and carotid plaque characteristics were found in the studied groups (Table 3).

### 3.5. ROC Analysis

Following the ROC analysis, SU levels yielded a satisfactory performance, with AUCs of 0.65 (95% CI 0.50–0.73), 0.62 (95% CI 0.55–0.68) and 0.68 (95% CI 0.59–0.76) in detecting subjects with increased IMT, with carotid plaques and vulnerable plaque types, respectively, only within the group of central obesity (Figure 3). For all other obesity groups, the performance of AUC was unsatisfactory, ranging between 0.5 and 0.6 (Figure 3).

## 4. Discussion

In this study, we investigated the levels of SU, markers of carotid atherosclerosis as well as their associations in subjects with different types of obesity. Specifically, we demonstrated that SU levels and IMT values were the highest among subjects with central and mixed types of obesity. However, higher levels of SU were associated with increased IMT occurrence and were able to detect subjects with carotid atherosclerosis only within the group of central obesity. These findings point to the relevance of SU levels for carotid atherosclerosis in subjects with obesity, particularly in those with central obesity.

A substantial body of evidence points to a clear relation between high levels of SU and carotid atherosclerosis. Previous studies have shown a positive correlation between SU levels and carotid IMT values [21,24,37], documented in healthy subjects as well as in subjects with hypertension, diabetes, metabolic syndrome and kidney diseases [3,21]. In our study, we found that higher SU levels were associated with a higher occurrence of increased IMT but only among subjects with central obesity. Moreover, only within the group of central obesity, subjects with high SU levels displayed higher estimates of IMT compared to obese subjects with normal SU levels. At the same time, we did not find a significant association between SU levels and the occurrence of increased IMT in subjects with general obesity, despite the reports on tight correlation between SU and BMI [8,9,10]. This might be related to a relatively small number of subjects with general obesity included in our study or to the fact that we included only obese subjects, in contrast to other studies that included subjects with normal and high BMI values. In addition, this might be attributed to the evidence that values of WC correlate better than BMI with subclinical atherosclerosis, as evaluated by IMT [38]. Taking into account the findings of previous studies claiming that central obesity is one of the main contributors of increased SU levels in the general population [5,13] and our findings of the highest SU levels and IMT values among subjects with central obesity, it might be postulated that high SU levels favor carotid atherosclerosis, particularly in subjects with central obesity.

Several studies have explored the link between SU levels and the occurrence of carotid plaques, with otherwise inconsistent findings; some studies found no association between SU and the presence of carotid plaques [39], while other studies attested a clear association [22,40,41,42]. According to the latter, the association between SU and carotid plaques increased with elevating SU levels and demonstrated a dose-response relation only in men [22] or only in women [41]. Interestingly, SU was shown to be an independent risk factor for incidence of carotid plaques only in men without metabolic syndrome and not in men or women with metabolic syndrome [40]. Based on our findings, we could not find any positive association between SU levels and the presence of carotid plaques in either obesity group. A higher number and area of plaques could be observed in patients with central and mixed types of obesity, but without reaching statistical significance. On the other side, an inverse association between SU levels and the occurrence of carotid plaques was observed in non-obese subjects. One may speculate that this might be related to the beneficial effects of SU, which exerts not only pro-oxidative and pro-inflammatory activity but also has an extracellular anti-oxidative activity [43,44]. Recent studies suggested that SU levels are also related to carotid plaque instability [23,42]. Thus, elevated SU levels were independently associated with the prevalence of vulnerable carotid plaques in middle-aged adults [23] and were more commonly detected in symptomatic compared to asymptomatic carotid plaques [42], thereby indicating that SU may play an important role in carotid plaque inflammation and determine plaque vulnerability and subsequent rupture. Our results did not find any association between SU levels and the presence of vulnerable plaques in patients with obesity. A higher frequency of vulnerable plaques was observed in subjects with central obesity compared to subjects with general and mixed obesity types (63% vs. 53% vs. 58%, respectively); however, this was not statistically significant. In this regard, the heterogeneity among the available studies in terms of subjects’ age, concomitant comorbidities, confounding factors and used measures of obesity may account for existing controversies.

The clinical relevance of studying SU levels relies on their strong correlation with cardiovascular risk [3,45]. There is emerging evidence that an increase in SU to levels even lower than those typically associated with gout is independently associated with increased cardiovascular risk in both the general and hypertensive populations [46]. Similar studies should also be conducted in other high risk populations, including subjects with obesity. The latest guidelines on arterial hypertension (2018, European Society of Cardiology/European Society of Hypertension ESC/ESH) have officially introduced the evaluation of SU for the stratification of a patient’s cardiovascular risk [46]. Measurement of SU levels is recommended as part of the screening of hypertensive patients [3]. Our observations on the ability of SU levels to detect subjects with increased IMT and subjects with vulnerable plaques within the group of central obesity highlights the necessity of careful SU management in this specific population. The control of SU levels as an early prevention strategy might be helpful in reducing the risk of atherosclerosis, and hence reduce cardiovascular and cerebrovascular diseases. In this regard, pharmacological interventions, e.g., with sodium/glucose co-transporter-2 (SGLT2) inhibitors, which are antidiabetic agents that also lower SU levels, may be an important component of this strategy [44].

Several limitations are applicable to our study. First, data on concomitant administration of drugs, other than aspirin, affecting the SU levels (e.g., thiazide diuretics, nicotinic acid, etc.) were incomplete for the included subjects, and their effects could not be assessed. Second, subjects with general obesity were relatively underrepresented in our study, with only 59 subjects included. Future studies with a larger sample size may provide additional findings regarding the SU effects on carotid atherosclerosis in this patient population. Third, categorization of subjects into general, central and mixed obesity types is based on anthropometric measures and does not take into account physiological or pathological metabolic characteristics. Obesity phenotypes are heterogeneous, ranging from metabolically healthy to unhealthy obesity, with different risks for cardiovascular diseases [47]. Fourth, although our study revealed the association between SU levels and IMT, no causality can be inferred due to the cross-sectional nature of the study. Finally, even though the performance of ROC models was satisfactory, increased levels of SU cannot be considered a reliable marker in detecting subjects with carotid atherosclerosis, thus requiring further investigation and validation studies.

## 5. Conclusions

The results of our study indicate that among studied obesity types, subjects with central and mixed phenotypes were characterized by the highest SU and IMT values. Only in subjects with central obesity, higher SU levels were significantly related to a higher occurrence of increased IMT and were able to detect those with carotid atherosclerosis. Therefore, subjects with central obesity require a close observation for managing SU levels and its related complications.

## Figures and Tables

**Figure 1 biomedicines-11-01897-f001:**
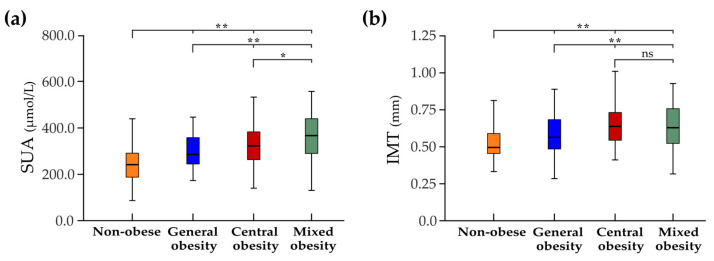
Serum urate (SU) and intima-media thickness (IMT) across the studied groups. Comparison of SU levels (**a**) and IMT values (**b**) between the non-obese and subjects with general, central and mixed obesity; * *p* < 0.05, ** *p* < 0.001, ns—non-significant.

**Figure 2 biomedicines-11-01897-f002:**
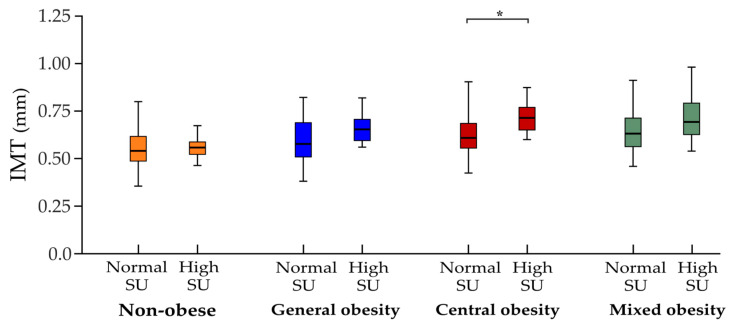
Estimates of intima-media thickness (IMT) across subject subgroups. Comparison of IMT values between subjects with normal and high serum urate (SU) levels among the non-obese and general, central and mixed obesity groups; * *p* < 0.05.

**Figure 3 biomedicines-11-01897-f003:**
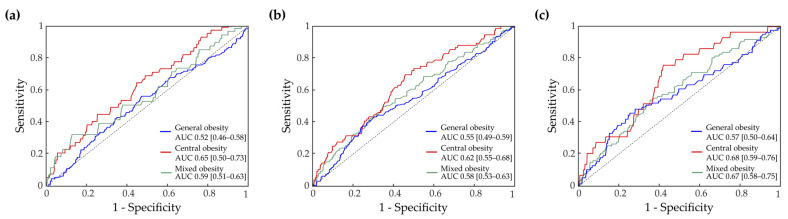
Receiver operating characteristic (ROC) curve analysis. Area under the ROC curve (AUC) assessing the performance of serum urate in detection of subjects with increased IMT (**a**), subjects with carotid plaques (**b**) and subjects with vulnerable carotid plaques (types 1 and 2) (**c**) in each obesity group. In the legend, AUCs with 95% confidence intervals are shown for each obesity group.

**Table 1 biomedicines-11-01897-t001:** Demographic, clinical and laboratory characteristics of the study subjects.

Parameter	Non-Obese (n = 686)	General Obesity (n = 59)	Central Obesity (n = 616)	Mixed Obesity (n = 715)	*p*-Value
Age, mean ± SD	42.8 ± 13.3	51.5 ± 10.4	48.9 ± 13.0	52.4 ± 10.9	**<0.001 ^a^ **
Sex (female/male), n (%)	602 (88)/84 (12)	58 (98)/1 (2)	169 (27)/447 (73)	478 (67)/237 (33)	**<0.001 ^b^ **
Systolic BP, mean ± SD, mmHg	125.8 ± 17.2	139.1 ± 19.1	137.0 ± 20.2	145.7 ± 21.3	**<0.001 ^c^ **
Diastolic BP, mean ± SD, mmHg	79.0 ± 10.0	84.5 ± 10.2	83.5 ± 10.6	88.3 ± 10.8	**<0.001 ^c^ **
BMI, mean ± SD, kg/m^2^	23.8 ± 3.0	31.9 ± 1.7	26.3 ± 2.4	34.9 ± 4.1	NA
Abdominal circumference, mean ± SD, cm	80.9 ± 7.8	89.1 ± 5.8	96.1 ± 7.9	109.6 ± 9.5	NA
Arterial hypertension, n (%)	113 (16)	26 (44)	164 (27)	373 (52)	**<0.001 ^b^ **
Diabetes mellitus, n (%)	12 (7)	5 (8)	25 (4)	80 (11)	**<0.001 ^b^ **
Dyslipidemia, n (%)	377 (55)	45 (76)	411 (66)	544 (76)	**<0.001 ^b^ **
Smoking, n (%)	49 (7)	1 (2)	159 (25)	60 (8)	**<0.001 ^b^ **
Serum urate, mean ± SD, μmol/L	234.2 ± 59.8	272.9 ± 61.8	301.2 ± 73.1	309.4 ± 82.2	**<0.001 ^d^ **
Increased serum urate, n (%)	12 (2)	6 (10)	48 (8)	126 (17)	**<0.001 ^b^ **
FBG, mean ± SD, mmol/L	4.7 ± 0.7	4.8 ± 0.6	5.0 ± 1.2	5.5 ± 1.8	**<0.001 ^d^ **
Creatinine, mean ± SD, mmol/L	68.6 ± 14.4	72.9 ± 10.8	76.5 ± 16.5	73.4 ± 16.1	**<0.01 ^e^ **
Triglycerides, mean ± SD, mmol/L	1.0 ± 0.6	1.4 ± 0.6	1.3 ± 0.7	1.5 ± 0.9	**<0.001 ^e^ **
Total-C, mean ± SD, mmol/L	5.0 ± 0.9	5.4 ± 1.0	5.3 ± 1.1	5.5 ± 1.0	**<0.001 ^e^ **
LDL-C, mean ± SD, mmol/L	3.1 ± 0.9	3.5 ± 1.1	3.7 ± 0.9	3.7 ± 0.9	**<0.001 ^e^ **
HDL-C, mean ± SD, mmol/L	1.6 ± 0.4	1.5 ± 0.4	1.4 ± 0.4	1.3 ± 0.3	**<0.001 ^e^ **
Concomitant medication (aspirin), n (%)	54 (8)	16 (27)	93 (15)	227 (32)	**<0.001 ^b^ **

Significant *p*-values are marked in bold. BMI: body mass index; BP: blood pressure; C: cholesterol; FBG: fasting blood glucose; HDL-C: high-density lipoprotein cholesterol; LDL-C: low-density lipoprotein cholesterol; NA: not applicable; SD: standard deviation. ^a^ One-way analysis of variance; ^b^ Chi-square test; ^c^ General linear models adjusted for age and sex; ^d^ General linear models adjusted for age, sex, concomitant medication (aspirin), creatinine and other cardiovascular risk factors (i.e., arterial hypertension, diabetes mellitus, smoking, total C, LDL-C, and HDL-C); ^e^ General linear models adjusted for age, sex, and other cardiovascular risk factors (i.e., arterial hypertension, diabetes mellitus, and smoking).

**Table 2 biomedicines-11-01897-t002:** Carotid ultrasound characteristics of the study subjects.

Parameter	Non-Obese (n = 686)	General Obesity (n = 59)	Central Obesity (n = 616)	Mixed Obesity (n = 715)	*p*-Value
IMT, mean ± SD, mm	0.57 ± 0.13	0.62 ± 0.14	0.67 ± 0.16	0.68 ± 0.16	**<0.001 ^a^ **
Increased IMT, n (%)	17 (2)	6 (10)	44 (7)	70 (10)	**<0.001 ^b^ **
Presence of plaques, n (%)	76 (11)	15 (25)	138 (22)	189 (26)	**<0.001 ^b^ **
Number of plaques, mean ± SD	0.21 ± 0.03	0.33 ± 0.11	0.45 ± 0.04	0.49 ± 0.03	**<0.001 ^a^ **
Area of plaques (mm^2^), mean ± SD	4.4 ± 15.7	5.6 ± 11.3	10.9 ± 27.3	11.0 ± 24.5	**<0.001 ^a^ **
Type of plaques:					0.77 ^b^
vulnerable (types 1, 2)	29 (38)	7 (47)	51 (37)	79 (42)	
stable (types 3, 4, 5)	47 (62)	8 (53)	87 (63)	110 (58)	

Significant *p*-values are marked in bold. IMT: intima-media thickness; SD: standard deviation. ^a^ General linear model adjusted for age, sex, concomitant medication (aspirin), creatinine and other cardiovascular risk factors (arterial hypertension, diabetes mellitus, smoking, total C, LDL-C and HDL-C); ^b^ Chi-square test.

**Table 3 biomedicines-11-01897-t003:** Results of the binary logistic regression analysis on the association between serum urate levels and carotid atherosclerosis.

Subject Group	OR	95% CI	*p*-Value
**Presence of increased IMT ^a^**			
Non-obese	0.992	0.985–1.001	0.06
General obesity	1.001	0.987–1.015	0.91
Central obesity	1.027	1.021–1.038	**<0.01**
Mixed obesity	1.001	0.984–1.019	0.89
**Presence of carotid plaques ^a^**			
Non-obese	0.992	0.988–0.996	**<0.001**
General obesity	1.002	0.992–1.011	0.74
Central obesity	1.005	0.993–1.009	0.70
Mixed obesity	1.008	0.996–1.014	0.88
**Presence of vulnerable carotid plaques ^a^**			
Non-obese	1.001	0.978–1.063	0.71
General obesity	1.004	0.995–1.015	0.90
Central obesity	0.997	0.990–1.005	0.11
Mixed obesity	0.999	0.997–1.001	0.41

Significant *p* values are marked in bold. CI: confidence interval; IMT: intima-media thickness; OR: odds ratio. ^a^ Models adjusted for age, sex, concomitant medication (aspirin), creatinine and other cardiovascular risk factors (arterial hypertension, diabetes mellitus, smoking, total C, LDL-C and HDL-C).

## Data Availability

The data are available from the corresponding author upon reasonable request.

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
