# Peer review of "Serum Urate Levels and Ultrasound Characteristics of Carotid Atherosclerosis across Obesity Phenotypes"

_biomedicines, 2023, doi:10.3390/biomedicines11071897_

Round 1
Reviewer 1 Report
This is a nicely conducted analysis taking advantage of large prospective study.
My comments are minor, but I’m not convinced that the 4 group (non, general, central and mixed) categories are the best way to analyze the data. The general group is so small, I don’t see it that helpful. The 4 categories combine 2 measures, BMI and waist circumference (WC). The following need not go into the manuscript, but I would like greater reassurance that more thorough bivariate explorations (SU vs. BMI and SU vs. WC) were conducted. Is dichotomization justified at the cut points (rather than continuous measures or other cut points). I appreciate the a priori cut points, but I would like this evaluated. I don’t like the very small general category. The authors might consider separate dichotomous (or continuous) variables (rather than 4 group). Then an interaction term could be considered. Again if general obesity in absence of central obesity is so infrequent, the modeling may not be helpful.
Is renal function available? Since that impacts SU and atherosclerosis, that would be important. Similarly is med list available? It would be nice to include diuretics. (More complicated so maybe it wont’ add clarity.)
Minor point, when you define groups by BMI and WC, I do not think you should include p-values (Table 1). I suggest deleting p-values for these rows.
Other minor point. Serum urate (SU) is the preferred term. Please see G-CAN statement on nomenclature. PMID: 31501138
Minor point. I found this wording confusing. No difference? (But there were and you state p<0.05) “No differences in systolic or diastolic BP were attested across 192 obesity types (all post-hoc p < 0.05).”
A few issues with this statement, “the cut-off values adopted in our and previous studies (i.e. >6 in women and >7 mg/dl in men) are based on the satura- 338 tion point of uric acid. Using lower cut-off values may turn more clinically relevant, since 339 existing evidence suggests that even lower levels of SUA have a negative impact on car- 340 diovascular risk [35].”
I understand using pre-specified cut offs, but if you express a concern and posit an alternative, you should look at that possibility. You are able to do that. Secondly men and women do not have different solubility levels, so the first part of the statement is inaccurate. 6 and 7 are based on solubility, as men have higher SU averages, a higher threshold has been selected. (I’m not sure this is well justified for most scientific hypotheses, but it is a standard.) I’m OK with the cut-offs, but state it is convention and that you did some sensitivity (e.g. changed women’s cutoff, looked at lower levels). Don’t just say it is problematic and leave it there. (Critique not meant to sound grumpy, apologies if it comes across that way)
Overall, excellent analysis and write up.
At discretion of authors, a single sentence on SLGT2 inhibitors might add future direction given pleiotropic effects target your variables of interest.
Reviewer 2 Report
The manuscript entitled Serum uric acid levels and ultrasound characteristics of carotid atherosclerosis across obesity phenotypes is an original article. The authors assessed the interrelations between SUA levels and carotid atherosclerosis (IMT and carotid plaque characteristics) in subjects with general, central and mixed (general-central) types of obesity.
In patients with central obesity, higher SUA levels were significantly related to a higher occurrence of increased IMT and were able to detect those with carotid atherosclerosis.
This is non-invasive, simple, interesting clinical study.
The article is very well written. Methodology is well detailed. The results are well presented.
Minor revision
I would suggest adding inclusion and exclusion criteria.
Thiazide and thiazide like diuretics may have as adverse effect hyperuricemia. It is mandatory putting in table 3 the drugs, which could interfere with SUA level.
Table 3: How do you explain significant p for presence of carotid plaques in non-obese patients?
